# Outcomes of COVID-19 Hospitalized Patients Previously Treated with Renin-Angiotensin System Inhibitors

**DOI:** 10.3390/jcm9113472

**Published:** 2020-10-28

**Authors:** Elena-Mihaela Cordeanu, Lucas Jambert, Francois Severac, Hélène Lambach, Jonathan Tousch, Marie Heitz, Corina Mirea, Amer Hamadé, Waël Younes, Anne-Sophie Frantz, Hamid Merdji, Valérie Schini-Kerth, Pascal Bilbault, Ferhat Meziani, Patrick Ohlmann, Emmanuel Andres, Dominique Stephan

**Affiliations:** 1Department of Hypertension, Vascular Disease and Clinical Pharmacology, Strasbourg Regional University Hospital, 67091 Strasbourg, France; helene.lambach@chru-strasbourg.fr (H.L.); jonathan.tousch@chru-strasbourg.fr (J.T.); marie.heitz2@chru-strasbourg.fr (M.H.); corina.mirea@chru-strasbourg.fr (C.M.); anne-sophie.frantz@chru-strasbourg.fr (A.-S.F.); Dominique.stephan@chru-strasbourg.fr (D.S.); 2Department of Vascular Medicine, Mulhouse Regional Hospital, 68100 Mulhouse, France; jambertlucas@gmail.com (L.J.); hamadea@ghrmsa.fr (A.H.); 3Division of Public Health, Methodology and Biostatistics, University Hospitals of Strasbourg, 67091 Strasbourg, France; francois.severac@chru-strasbourg.fr; 4Department of Vascular Medicine, Colmar Regional Hospital, 68000 Colmar, France; wael.younes@ch-colmar.fr; 5Intensive care and Reanimation Department, Strasbourg Regional University Hospital, 67091 Strasbourg, France; hamid.merdji@chru-strasbourg.fr (H.M.); ferhat.meziani@chru-strasbourg.fr (F.M.); 6UMR 1260 INSERM Regenerative Nanomedecine, Faculty of Pharmacy, Strasbourg University, 67400 Illkirch, France; valerie.schini-kerth@unistra.fr; 7Emergency Department, Strasbourg Regional University Hospital, 67091 Strasbourg, France; pascal.bilbault@chru-strasbourg.fr; 8Cardiology Department, Strasbourg Regional University Hospital, 67091 Strasbourg, France; patrick.ohlmann@chru-strasbourg.fr; 9Internal Medicine Department, Strasbourg Regional University Hospital, 67091 Strasbourg, France; emmanuel.andres@chru-strasbourg.fr

**Keywords:** COVID-19, SARS-CoV-2, renin–angiotensin system inhibitor, angiotensin-converting enzyme, angiotensin II receptor blocker, propensity score

## Abstract

(1) Background: Severe acute respiratory syndrome coronavirus 2 (SARS-CoV-2) penetrates respiratory epithelium through angiotensin-converting enzyme-2 binding, raising concerns about the potentially harmful effects of renin–angiotensin system inhibitors (RASi) on Human Coronavirus Disease 2019 (COVID-19) evolution. This study aimed to provide insight into the impact of RASi on SARS-CoV-2 outcomes in patients hospitalized for COVID-19. (2) Methods: This was a retrospective analysis of hospitalized adult patients with SARS-CoV-2 infection admitted to a university hospital in France. The observation period ended at hospital discharge. (3) Results: During the study period, 943 COVID-19 patients were admitted to our institution, of whom 772 were included in this analysis. Among them, 431 (55.8%) had previously known hypertension. The median age was 68 (56–79) years. Overall, 220 (28.5%) patients were placed under mechanical ventilation and 173 (22.4%) died. According to previous exposure to RASi, we defined two groups, namely, “RASi” (*n* = 282) and “RASi-free” (*n* = 490). Severe pneumonia (defined as leading to death and/or requiring intubation, high-flow nasal oxygen, noninvasive ventilation, and/or oxygen flow at a rate of ≥5 L/min) and death occurred more frequently in RASi-treated patients (64% versus 53% and 29% versus 19%, respectively). However, in a propensity score-matched cohort derived from the overall population, neither death (hazard ratio (HR) 0.93 (95% confidence interval (CI) 0.57–1.50), *p* = 0.76) nor severe pneumonia (HR 1.03 (95%CI 0.73–1.44), *p* = 0.85) were associated with RASi therapy. (4) Conclusion: Our study showed no correlation between previous RASi treatment and death or severe COVID-19 pneumonia after adjustment for confounders.

## 1. Introduction

Human Coronavirus Disease 2019 (COVID-19) resulting from a newly described respiratory viral infection with severe acute respiratory syndrome coronavirus 2 (SARS-CoV-2) originally started in December 2019 in Wuhan, China, and rapidly became a global pandemic, as officially recognized by the World Health Organization on the 11 March 2020. A minor proportion of infected individuals (15%) develop severe forms of infection requiring hospitalization, while 5% are critical and need intensive care support and mechanical ventilation [1]. Cardiovascular risk factors, such as hypertension, diabetes, and obesity, as well as cardiovascular disease, are associated with worse prognosis [2]. Renin–angiotensin system (RAS) inhibitors (RASi) represent a first-line antihypertensive drug class, largely prescribed in hypertensive, diabetic, and heart failure patients in virtue of their long-term cardiovascular and renal protective effects. Similar to SARS-CoV, but with higher receptor affinity, SARS-CoV-2 penetrates respiratory epithelium through the binding of its spike envelope protein to cell membrane angiotensin-converting enzyme-2 (ACE2) [3,4]. Physiologically, ACE2 signaling through the MAS/G-coupled protein receptor pathway possesses a cardiovascular protective function, balancing the effect of RAS activation [5]. Contrary to ACE, ACE2 activity is not directly regulated by RASi but seems to be upregulated by high levels of angiotensin I, a consequence of RASi administration [6]. As ACE2, which is highly expressed in the lungs, the kidneys, the gut, and the brain, plays a key role in viral cell entry, the implication of RAS and RASi in the severity of COVID-19 infection is being questioned [7]. We report herein a retrospective analysis of adult hospitalized patients from a university hospital from the Eastern France, one of the areas in Europe most affected by the first wave of the COVID-19 pandemic.

## 2. Experimental Section

### 2.1. Study Design and Patient Selection

We performed a retrospective analysis of electronic medical records of hospitalized COVID-19 patients admitted to the University Hospital of Strasbourg between 25 February 2020 (date of admission of the first case) and 1 April 2020. The study was approved by the Strasbourg University Hospital Ethical Committee. All patients aged more than 18 years old were selected on the basis of laboratory-confirmed COVID-19 infection by positive reverse-transcriptase polymerase chain reaction (RT-PCR). A local RT-PCR kit was used to detect SARS-CoV-2. Patients hospitalized for less than 24 h in the emergency room or those with asymptomatic SARS-CoV-2 infection were excluded. Thus, all patients for whom discharge status was known, i.e., either death during hospitalization or survival to discharge, were included. The observation period ended at discharge.

### 2.2. Baseline Variables

Data concerning medical history, chronic medication, clinical presentation, laboratory findings, and low-dose pulmonary computed tomography (CT) lesions were collected. Arterial hypertension and RASi treatment were collated according to patient history, as well as their continuation during hospitalization. Antiviral, antibiotic, and anticoagulant treatment during hospitalization were equally reported.

### 2.3. Outcome Assessment

For the purpose of this study, the observation period ended at hospital discharge, with a median length of stay of 11 days (interquartile range (IQR) 6–21) and a median observation time from the first COVID-19 symptoms of 18 days (IQR 11–28). All patient data were collected during hospitalization. RASi treatment was noted when patients received either angiotensin-converting enzyme inhibitors (ACEIs) or angiotensin II receptor blockers (ARBs), or both. The main evaluation criterion was death from any cause. Secondary evaluation criterion was “severe COVID-19 pneumonia” defined by at least one of the following criteria: (1) Leading to death, and/or requiring (2) oxygen flow at a rate of at least 5 L/min, and/or (3) high-flow nasal oxygen (HFNO) therapy, and/or (4) noninvasive ventilation (NIV), and/or (5) orotracheal intubation (OTI). A comparison between angiotensin-converting enzyme inhibitors (ACEIs) and angiotensin II receptor blockers (ARBs) on all-cause mortality, severe pneumonia, and acute renal impairment was also performed. Severe sepsis, secondary bacterial infection, venous thromboembolism, stroke, atrial fibrillation, liver injury, and major bleeding during hospitalization were noted. The evaluation criteria were adjudicated by senior physicians of the vascular medicine unit.

### 2.4. Statistical Analysis

This was a retrospective cohort study, therefore, no power calculation was performed. Continuous variables were expressed as mean ± standard deviation (SD) or median with interquartile range (IQR), depending on their distribution. The normality of the distribution was assessed using the Shapiro–Wilk test. Categorical variables were presented as numbers of cases (percentages). Patients were divided according to previous RASi treatment (RASi or RASi-free). The association between several baseline factors and the risk of death was assessed by univariate analysis. Clinically pertinent risk factors associated with mortality with univariate tests considered significant were selected as candidates for the multivariate logistic regression analysis. The risk of death or severe pneumonia associated with RASi was assessed and results were expressed as odds ratios (ORs) with 95% confidence intervals (CI). In order to address potential sources of bias and further compare outcomes between RASi users and nonusers, we performed a propensity score analysis. Propensity scores were generated using an outcome-blinded logistic regression of RASi treatment on the baseline covariates considered to be influential for the use of RASi, namely, age (categorized as a continuous variable), gender, high blood pressure, diabetes, dyslipidemia, obesity, smoking, chronic kidney disease, chronic heart disease, reduced ejection fraction, cognitive impairment, previous venous thromboembolism, antithrombotic treatment on admission, diuretics on admission, and beta-blockers on admission. The Kalplan–Meier estimator was employed to compute survival curves over the observation period. A Cox proportional hazard model adjusted using the propensity score was performed to compare the risk of death and the composite of death and severe pneumonia. Results were expressed as hazard ratios (HRs) with 95% confidence intervals. A *p*-value of <0.05 was considered statistically significant. All analyses were performed using R software version 3.2.2 (www.r-project.org).

## 3. Results

### 3.1. Patients Characteristics at Baseline

A total of 943 COVID-19 patients were admitted to the University Hospital of Strasbourg from 25 February 2020 to 1 April 2020, of whom 772 (57.5% of males, mean age of 66.4 ± 16.8 ranging from 19 to 100 years) were included in this analysis, after exclusion of patients hospitalized for less than 24 h (*n* = 145), minors (*n* = 14), and patients hospitalized for other medical reasons and incidentally found positive for SARS-CoV-2 PCR (*n* = 12) (Figure 1). Among the included individuals, 431 (55.8%) patients had previously known high blood pressure (HBP) and 282 (36.5%) were treated with an RASi (129 received an ACEI, 152 received an ARB, and 1 patient received an ACEI + ARB). Fever (83%), fatigue (72%), cough (71%), and dyspnea (69%) were the most frequent symptoms. The cohort was divided into two subgroups based on previous treatment with ACEIs/ARBs, namely, “RASi” (*n* = 282) and “RASi-free” (*n* = 490). Both groups exhibited similar clinical presentations and similar time delays between first symptoms and hospital admission (data not shown). Patients from the RASi group were older, had higher cardiovascular risk profiles, and were more frequently victims of cardiovascular disease (CVD) or chronic kidney disease (CKD). Biological marker severity (lymphocyte count, C-reactive protein (CRP), and D-dimer count) and CT scan extension were comparable between groups (Table 1).

In order to obtain comparable populations of RASi-exposed and -unexposed subjects, propensity score-adjusted analyses were performed for 226 patients selected on the basis of covariates of adjustment deemed significant for RASi prescription (the adjustment variables are listed in Section 2.4.). Baseline characteristics of the propensity score (PS)-matched cohort are detailed in Table 1; no significant differences were observed between RASi-treated patients and RASi-free patients.

### 3.2. In-Hospital Outcomes

Overall, 220 (28.5%) patients were placed under mechanical ventilation (28.4% in the RASi group versus 28.6% in the RASi-free group), of whom 71 (32%) died (36.2% in the RASi group versus 30% in the RASi-free group). All-cause mortality was 22.4% (*n* = 173). Patients from the RASi group had overall higher oxygen therapy necessities but equivalent recourse to high-flow nasal oxygen (HNFO), noninvasive ventilation (NIV), or orotracheal intubation (OTI). Patients treated with RASi had higher in-hospital mortality than those not receiving RASi (29.1% versus 18.6%). A detailed description of in-hospital complications is shown in Table 2.

### 3.3. Survivors Versus Nonsurvivors

A comparison between survivors and nonsurvivors is shown in Table 3. A total of 125 patients continued RASi treatment during hospitalization in similar proportions among survivors and nonsurvivors. No other differences in terms of in-hospital therapy (anticoagulation, antiviral, antibiotics) were noted between survivors and nonsurvivors. In univariate analysis, mortality was associated with higher age, conventional cardiovascular risk factors (hypertension, type 2 diabetes, dyslipidemia, smoking), active cancer, chronic kidney disease (CKD), ischemic heart disease, previous antithrombotic therapy, RASi treatment, lymphopenia, and elevation of CRP and D-dimer count. Kaplan–Meier unadjusted survival curves showed an HR of 1.52 ((CI95% 1.20–2.03), *p* = 0.0007) for death of any cause when previously treated with an RASi (Figure 2A). In a multivariate logistic-regression model, age greater than 65 years old (OR 5.99, (95%CI 3.42–11.05)), active cancer (OR 2.87, (95%CI 1.51–5.43)), CKD (OR 2.96, (95%CI 1.79–4.89)), and previous antithrombotic treatment (OR 1.67 (95%CI 1.04–2.67)) were independently associated with death. Thus, RASi treatment and cardiovascular risk factors, except for age, were codependent variables (Appendix A). Similarly, in the propensity score-matched population, no significant difference was noted for all-cause death between groups (HR 0.93 (CI95% 0.57–1.50), *p* = 0.76) (Figure 2B).

### 3.4. Mild Versus Severe Forms of COVID-19 Infection

In univariate analysis, severe forms of COVID-19 infection (as defined in the methodology section) were associated with age of more than 65 years (odds ratio (OR) 1.51 (95%CI 1.12–2.02), *p* = 0.005), male gender (OR 2.21 (95%CI 1.66–2.97), *p* < 0.001), arterial hypertension (OR 1.71 (95%CI 1.25–2.29), *p* < 0.001), diabetes (OR 1.51 (95%CI 1.09–2.09), *p* = 0.012), obesity (OR 1.44 (95%CI 1.05–2.00), *p* = 0.024), previous RASi treatment (OR 1.54 (95%CI 1.14–2.09), *p* = 0.004), low lymphocyte count, i.e., <1000/µL (OR 2.43 (95%CI 1.75–3.39), *p* < 0.001), CRP ≥ 100 mg/L (OR 7.78 (CI95% 5.58–10.97), *p* < 0.001), D-dimer count ≥ 1500 µg/L (OR 8.94 (95%CI 5.20–15.71), *p* < 0.001), and troponin I ≥ 100 ng/L (OR 3.12 (95%CI 1.60–6.69), *p* = 0.001). Kaplan–Meier unadjusted event-free survival curves showed an HR of 1.20 ((95%CI 1.06–1.35), *p* = 0.0032) for severe forms of COVID-19 infection when previously treated with an RASi (Figure 3A). In a multivariate logistic-regression model, age greater than 65 years old (OR 1.78 (95%CI 1.15–2.74), *p* = 0.009), male gender (OR 1.58 (95%CI 1.10–2.28), *p* = 0.012), CRP ≥ 100 mg/L (OR 4.76 [95%CI 3.27–6.97], *p* < 0.001), and D-dimer count ≥ 1500 µg/L (OR 4.24 (95%CI 2.30–7.90), *p* < 0.001) were independently associated with infection severity. Thus, RASi treatment, as well as cardiovascular risk factors, were codependent variables (Appendix A). In the propensity score-matched population, no significant difference was noted for severe pneumonia between groups (HR 1.03 (95%CI 0.73–1.44), *p* = 0.85) (Figure 3B).

### 3.5. ACEIs Versus ARBs and Poor Outcomes

The interactions of ACEI (*n* = 129) or ARB (*n* = 152) use with mortality, risk of severe COVID-19 infection, and acute renal impairment were separately analyzed. One patient receiving both ACEI and ARB was excluded from this analysis. There was no difference observed between the two pharmacological classes concerning these outcomes, with a tendency toward a protective effect of ACEIs on renal function (Table 4).

## 4. Discussion

### 4.1. Main Results in Brief

Our study showed higher crude hazard ratios for death and severe pneumonia in RASi users. However, no correlation between RASi treatment and all-cause death or severe COVID-19 pneumonia was observed after adjustment for demographic and biological confounders in a multivariate regression model as well in a propensity score-matched population. No differences were noted between ACEIs and ARBs for the same evaluation criteria in the overall cohort. Furthermore, in-hospital continuation of ACEIs/ARBs was not associated with poorer outcomes.

Our findings were consistent with recent publications and guidelines recommending against discontinuation of RASi in COVID-19 infection. To date, several large-scale published retrospective cohorts found no negative impact of RASi on the COVID-19 clinical course. In an Italian population-based study including 6272 COVID-19 patients and 30,759 matched controls, Mancia et al. found no association between RASi and severe or fatal course of the disease, but did show that the use of RASi was more common among infected patients [8]. More recently, the ITA-COVID-19 RAS inhibitor group published a large-scale study of over 40,000 hospitalized patients showing no significant difference in mortality between RASi and other antihypertensive drugs, but a slightly higher mortality compared to nonusers of antihypertensive medications [9]. Based on medical records from New York University, Reynolds et al. studied the relationship between infection severity and previous treatments in a cohort of 5894 patients, finding no correlation with RASi treatment in either the global population or in the hypertensive subgroup [10]. Moreover, Yang et al. suggested a beneficial effect of RASi compared to other drug classes in hypertensive COVID-19 patients through reduction of the inflammatory response [11]. Other authors found a mortality reduction in RASi-treated patients, with a recent meta-analysis reinforcing this trend [12,13,14]. To date, only one randomized controlled trial (BRACE CORONA trial) prospectively evaluated two strategies, i.e., temporary discontinuation versus continuation of RASi during hospitalization of COVID-19 patients, finding no difference in terms of in-hospital and 30 day-mortality between groups [15]. Although some data convey the potential of RASi to increase the risk of SARS-CoV-2 infection, our study was not designed to analyze the infectivity risk in RASi-treated patients [8,10,16]. On the contrary, based on previous considerations concerning the detrimental role of angiotensin II (AngII) in acute respiratory distress syndrome, Rossi et al. suggested that high plasma levels of ACE2 could have protective effects, thus capturing the viral spike protein and preventing virus cell entry [17].

### 4.2. Hypothetical Pathogenic Mechanisms of RASi Impact on COVID-19 Evolution

At the beginning of the pandemic, conflicting evidence concerned the RASi effect on SARS-COV-2 infectivity and poor outcomes, leading to initial withholding of RASi administration in infected patients. Pathogenic mechanisms suggesting that RASi could have a potentially harmful effect on COVID-19 progression are based on the observation that high levels of angiotensin II (Ang II) were observed in COVID-19 patients and correlated with lung injury, supporting RAS activation during SARS-CoV-2 infection [18]. SARS-CoV-2 shares 82% genomic identity with 2002/2003 SARS-CoV, known for its deleterious vasoconstrictive, proinflammatory, and profibrotic effects, which was associated with RAS activation, [19]. Similar to SARS-CoV, ACE2, a type I transmembrane protein, is considered to be the key host cellular receptor for the SARS-CoV-2 spike [7,20,21,22]. ACEIs/ARBs were previously shown to indirectly upregulate ACE2 expression, which could expose individuals to COVID-19 infection. In vitro, ACE2 levels were correlated to SARS-CoV susceptibility, with some indirect proof suggesting that ACE2 expression might be correlated with human susceptibility to SARS-CoV-2 infection and severity. Furthermore, animal studies showed that SARS-CoV infection downregulated lung ACE2 expression causing lung injury, supporting a deleterious effect of ACE2 in COVID-19 infection [23]. Moreover, ACE2 expression may vary according to certain pathologic conditions and increase individual susceptibility to infection and progression to severe forms. Chronic heart disease individuals were previously shown to exhibit higher ACE2 expression levels, suggesting higher susceptibility to develop more severe forms of SARS-CoV-2 infection [24]. Age, sex, ans genetic variants do not seem to influence ACE2 expression [25,26,27]. On the other hand, ACE2 is the major enzyme catalyzing the conversion of AngII to Ang 1–7, which is known to lead to vasodilatation and vasoprotective effects through MAS receptor binding, thus making ACE2 a potent negative regulator of the renin–angiotensin system [21]. Thus, ACEIs/ARBs could play a double role in COVID-19 by increasing susceptibility to infection and alleviating acute lung injury [23].

### 4.3. Poor Prognosis Risk Factors

Our data confirmed previous reports on the correlation between older age as a major independent predictor of worse prognosis. No gender differences were observed in our cohort in terms of mortality. However, male subjects exhibited more severe forms of COVID-19 after adjustment for confounders. Indeed, some authors speculated that gender disparities in SARS-CoV-2 infection are related to hormone-modulation of ACE2 expression [28].

In our study, hypertension was associated with all-cause death in univariate analysis but not after adjustment for confounders, in line with Morales et al.’s results showing no increased risk of COVID-19 hospitalization for pneumonia in hypertensive patients treated with ACEIs or ARBs [29].

Compared to previous reports, our study showed a higher prevalence of hypertension (56% versus 15–30%) and similar frequency of ischemic heart disease (12% versus 15%) among COVID-19 patients [2,23]. According to Guan et al., hypertension was present in 23.7% of COVID-19 cases, while Zahng et al. found a prevalence of 30% [30,31]. In a Danish population-based medical database study, Christiansen et al. found prevalences of 13% and 26% for coronary artery disease and heart failure, respectively, in COVID-19 patients [32].

Our data showed that cardiovascular disease (CVD) was more frequent in patients with more severe forms of SARS-CoV-2 infection, which could be explained by ACE2 involvement; CVD is associated with higher age and CV risk factor prevalence, thereby triggering RAS activation with ACE/ACE2 disequilibrium alongside downregulation of ACE2 by SARS-CoV-2 infection. Both mechanisms lead to critically low levels of ACE2 and exaggerate AngII signaling in infected CVD patients, contributing to the severity of the disease through lung and cardiac injury [33,34].

According to AlGhatrif et al., age is a key factor regulating ACE2 expression and its implication in COVID-19, with older patients with CVD exhibiting lower ACE2 levels and higher RAS signaling, while younger patients without CVD show higher ACE2 levels and lower RAS signaling. Thus, older patients demonstrate lower disease incidence but higher severity compared to young individuals, who show higher incidence but lower severity [33].

In our study, infection severity, but not mortality, was associated with D-dimer and CRP elevation. Indeed, severe forms of COVID-19 are associated with particularly high inflammatory and prothrombotic responses, translating into the elevation of the acute-phase reactant C-reactive protein (CRP), low lymphocyte count, and elevation of the fibrin degradation product D-dimer [35,36]. According to Zhou et al., D-dimer elevation is the strongest predictor of mortality [37]. In a retrospective cohort of 247 hospitalized adults, Gomez et al. found no correlation between RAS treatment and poor outcomes, but a strong association between clinical worsening and neutrophil-to-lymphocyte ratio and D-dimer levels, underlining the prognostic value of inflammatory parameters [38]. Moreover, cardiac enzyme elevation, which indicates myocardial injury, was also correlated with mortality [39,40].

Concerning in-hospital continuation of ACEI/ARB treatment, we found no difference between survivors and nonsurvivors. Zhang et al. analyzed 1128 Chinese hypertensive patients, of whom 188 were treated with ACEIs/ARBs during hospitalization for COVID-19, and found a reduction in all-cause mortality in ACEI/ARB-treated patients after propensity score-matched analysis, with an adjusted HR of 0.37 (95% CI, 0.15–0.89) [41].

### 4.4. ACEIs Versus ARBs

No differences in all-cause death, severe pneumonia, or acute renal impairment were noted between ACEIs and ARBs in our study. In a cohort of 734 COVID-19 patients, Cheung et al. showed a reduced risk of severe disease in ACEI-treated patients, but not with ARB use (OR 0.14 (CI95% 0.02–0.87) versus 1.86 (CI95% 0.31–9.97)) [42]. This was consistent with the findings of Johnson et al., who showed that ACEI exposure was not associated with infection or death in a Medicare database study of more than 100,000 inhabitants, while ARB use increased infectivity rate but not mortality [43]. Contrarily, in a cohort of 1735 patients, Mehta et al. found a higher likelihood of intensive care unit (ICU) admission in patients taking ACEIs but not ARBs [16]. Such controversial effects may be explained by the different actions of ACEIs and ARBs on ACE2 levels; indeed, in animal studies, ARBs more frequently increased ACE2 expression while ACEIs seemed to exhibit less homogenous effects, with a tendency toward no effect [44].

### 4.5. Evidence in the Available Literature

Overall, several large observational cohorts and one randomized controlled study showed no association between RASi and severe disease or death [8,10,15]. Moreover, no relationship between RASi and the risk of contracting SARS-CoV-2 infection was formally proven, although remains suspected [16]. At this stage, given the rapid dissemination and severity of the pandemic, available data are almost exclusively derived from observational studies with heterogeneous populations, which are generally considered to be less robust. In order to validate conclusions drawn from these studies, replicability and consistency across studies are mandatory. As such, our study reinforces the currently published data in sustaining the unharmful impact of RASi treatment on COVID-19 outcomes, as stipulated by the latest guidelines of multiple scientific societies (European Society of Cardiology, European Society of Hypertension, American College of Cardiology) [44,45,46].

### 4.6. Limitations

This real-life observational study presented several limitations. Firstly, the exposed group had a higher cardiovascular risk profile and bias may have persisted in comparisons even after adjustment for confounders. Secondly, severity biological markers were not performed in all patients and at the same time on admission. Furthermore, our definition of the secondary evaluation criteria “severe COVID-19 pneumonia” may be criticized, as this includes deadly evolution and not only respiratory criteria. However, this was a strategic choice in order to avoid the exclusion of patients with a fatal outcome for whom, in regard to their comorbidities, ICU was deemed unreasonable. Finally, we deliberately excluded patients with negative RT-PCR results but positive CT scans in order to obtain a homogenous population.

## 5. Conclusions

Initial concerns raised about the safety of RASi in COVID-19 patients due to ACE2 involvement and high prevalence of cardiovascular comorbidities are now tempered by accumulating proof of their innocuity.

## Figures and Tables

**Figure 1 jcm-09-03472-f001:**
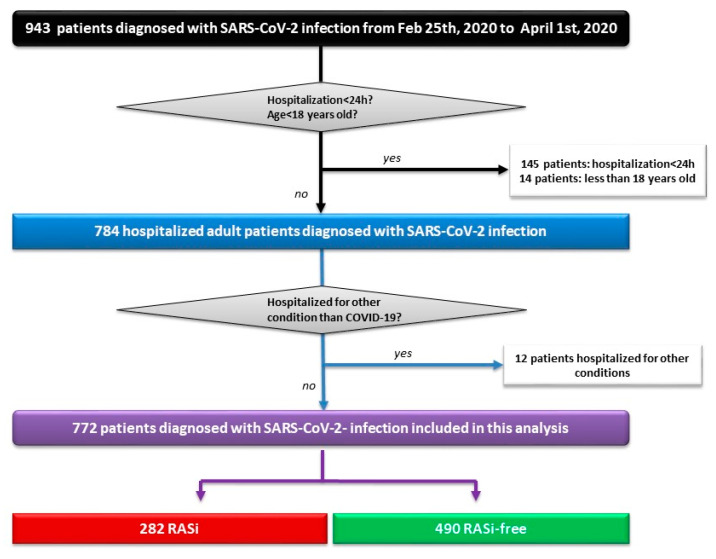
Study flowchart showing patient selection. h: Hours; Feb: February; RAS: Renin–angiotensin system; RASi: Renin–angiotensin system inhibitor(s).

**Figure 2 jcm-09-03472-f002:**
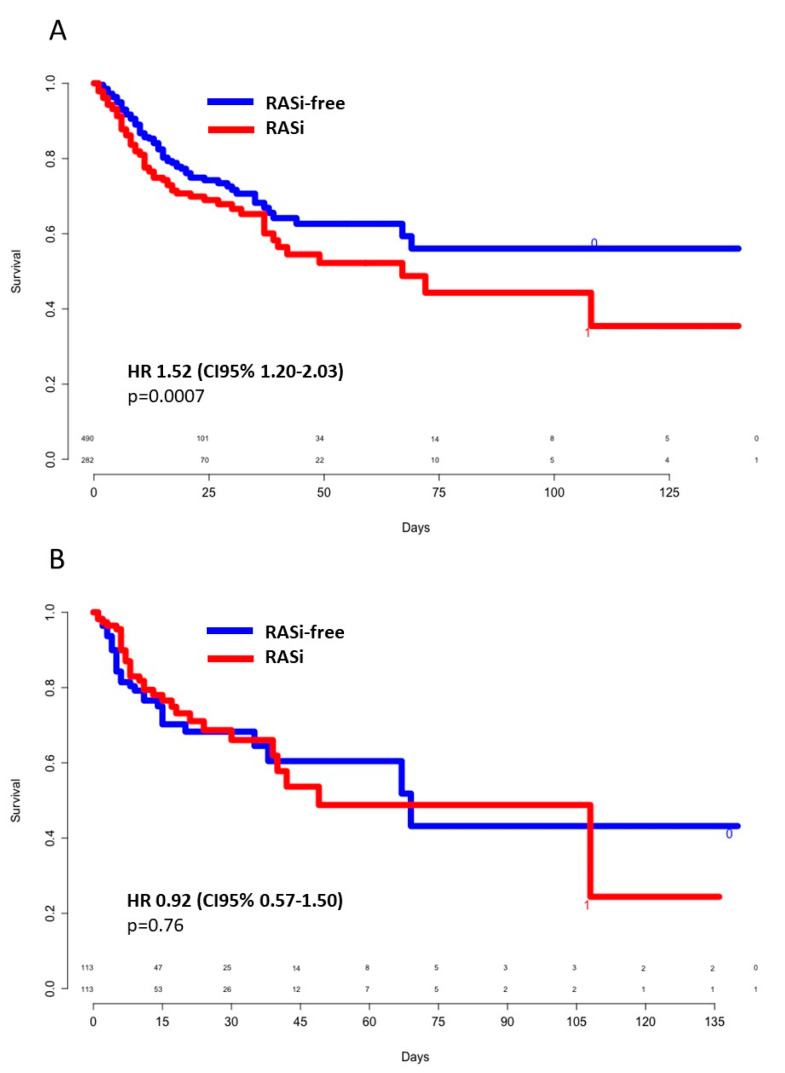
Crude (**A**) and propensity score-weighted (**B**) survival according to previous RASi use. CI: Confidence interval; HR: Hazard ratio; RASi: Renin–angiotensin system inhibitors.

**Figure 3 jcm-09-03472-f003:**
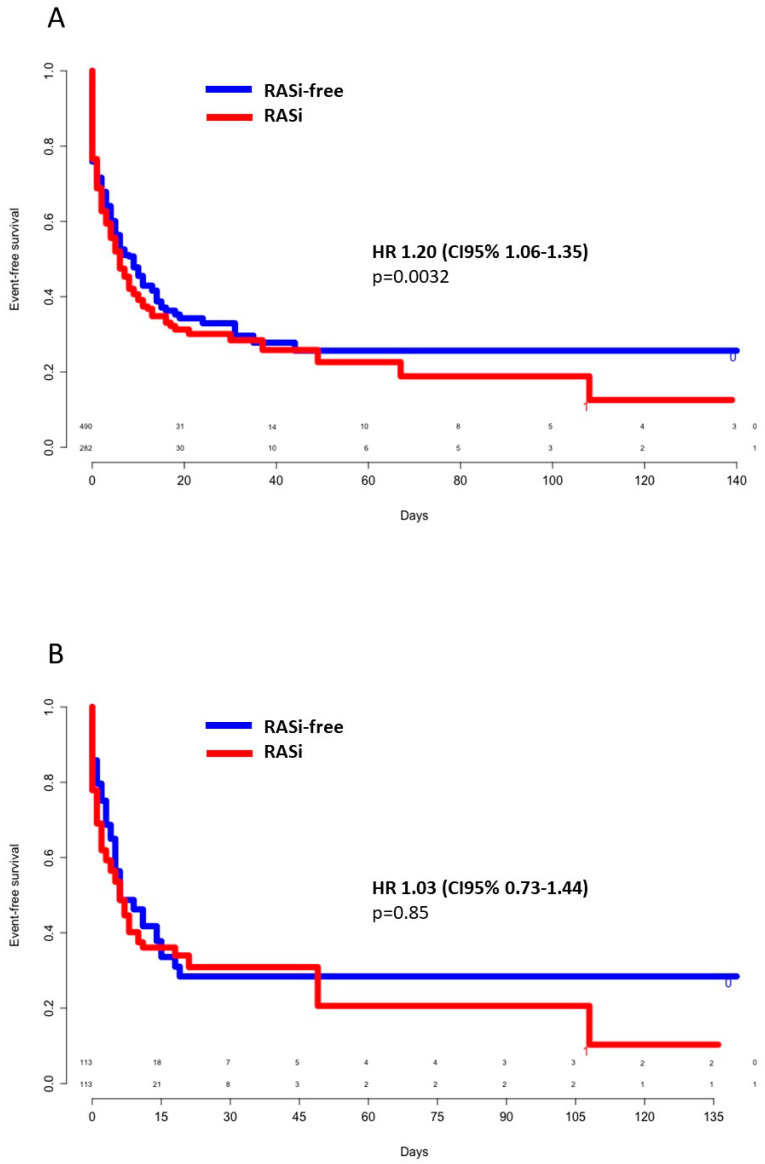
Crude (**A**) and propensity score-weighted (**B**) event-free survival for severe pneumonia according to previous RASi use. CI: Confidence interval; HR: Hazard ratio; RASi: Renin–angiotensin system inhibitors.

**Table 1 jcm-09-03472-t001:** Baseline demographics and clinical characteristics of the overall cohort and the propensity score-matched population according to previous RASi treatment.

	Overall Cohort			PS Cohort		
	RASi N(%)/M(IQR)	RASi-Free N(%)/M(IQR)	*p*-Value	RASi N(%)/M(IQR)	RASi-Free N(%)/M(IQR)	*p*-Value
**N**	**282**	**490**		**113**	**113**	
**Age (years)**	75 (66–83)	64 (50–75)	**<0.001**	73 (64–84)	73 (61–85)	0.61
Age ≥ 65 years old	221 (78.4)	243 (49.6)	**<0.001**	84 (74.3)	80 (70.8)	0.65
**Male**	170 (60.3)	274 (55.9)	0.27	68 (60.2)	61 (54)	0.42
**BMI (kg/m^2^) (N = 662)**	28 (25–33)	27 (24–31)	**0.004**	27 (24–31)	26 (24–31)	0.69
**eGFR (mL/min/1.73 m^2^) on admission** (N = 756)	68 (44–84)	88 (68.5–102.5)	**<0.001**	70 (44–84)	77 (43–93)	0.26
eGFR ≥ 90	45 (16)	231 (48.6)		22 (19.6)	35 (30.9)	0.07
60 ≤ eGFR < 90	122 (43.4)	143 (30.1)		44 (39.3)	32 (28.3)	0.11
30 ≤ eGFR < 60	76 (27)	66 (13.9)		32 (28.6)	27 (23.9)	0.51
eGFR < 30	38 (13.5)	35 (7.4)		14 (12.5)	19 (16.8)	0.47
**Cardiovascular risk factors**						
Hypertension (N = 769)	273 (96.8)	158 (32.4)	**<0.001**	104 (92)	104 (92)	1
Diabetes (N = 769)	134 (47.5)	85 (17.5)	**<0.001**	40 (35.4)	34 (30.1)	0.48
Dyslipidemia (N = 769)	166 (58.9)	107 (21.9)	**<0.001**	60 (53.1)	56 (49.6)	0.69
Smoking (history or current) (N = 671)	70 (27.9)	85 (20.2)	**0.029**	35 (31)	34 (30.1)	1
Obesity (N = 690)	107 (42.1)	134 (30.7)	**0.003**	35 (31)	35 (31)	1
**Medical history**						
Heart disease (N = 768)	83 (29.4)	41 (8.4)	**<0.001**	22 (19.5)	23 (20.3)	1
Ischemic heart disease	65 (23)	26 (5.3)	**<0.001**	14 (12.4)	12 (10.6)	0.83
Chronic heart failure	24 (8.5)	19 (3.9)	**0.011**	7 (6.2)	10 (8.8)	0.61
HFrEF	16 (5.7)	8 (1.6)	**0.004**	4 (3.5)	4 (3.5)	1
Chronic kidney disease (N = 769)	61 (21.6)	51 (10.5)	**<0.001**	26 (23)	31 (27.4)	0.54
Chronic respiratory disease (N = 769)	35 (12.4)	56 (11.5)	0.80	12 (10.6)	16 (14.2)	0.53
COPD	22 (7.8)	23 (4.7)	0.11	8 (7.1)	10 (8.9)	0.79
Active cancer	19 (6.7)	34 (6.9)	1	5 (4.4)	11 (9.7)	0.20
Cognitive impairment (N = 768)	48 (17.1)	47 (9.7)	**0.003**	18 (15.9)	20 (17.7)	0.85
VTE (N = 769)	27 (9.6)	31 (6.4)	0.138	13 (11.5)	14 (12.4)	1
**Admission treatment**						
**Antithrombotic treatment on admission**	159 (56.8)	108 (22.2)	**<0.001**	55 (48.7)	53 (46.9)	0.89
Antiplatelet (N = 766)	107 (38.4)	61 (12.5)	**<0.001**	40 (35.4)	28 (24.8)	0.11
Anticoagulation (N = 765)	61 (21.9)	57 (11.7)	**<0.001**	16 (14.2)	28 (24.8)	0.07
**Antihypertensive drugs** (N = 767)						
Diuretics	116 (41.4)	60 (12.3)	**<0.001**	42 (37.2)	41 (36.3)	1
Beta-blockers	121 (43.4)	94 (19.3)	**<0.001**	46 (40.7)	52 (46)	0.50
**COVID-19 diagnosis**						
Positive PCR	282 (100)	490 (100)	-	113 (100)	113 (100)	
Low-dose chest CT	230 (81.6)	388 (79.2)	0.42 *	91 (97.8)	81 (94.2)	0.38 *
normal	3 (1.3)	14 (3.6)		2 (2.2)	5 (5.8)	
uncertain abnormalities	5 (2.2)	7 (1.8)		2 (2.2)	3 (3.5)	
minimal abnormalities (<10%)	33 (14.3)	51 (13.1)		14 (15)	9 (10.5)	
moderate abnormalities (10–25%)	78 (33.9)	128 (33)		27 (29)	33 (38.4)	
important abnormalities (25–50%)	54 (23.5)	108 (27.8)		29 (31.2)	19 (22.1)	
severe abnormalities (50–75%)	43 (18.7)	65 (16.8)		16 (17.2)	14 (16.3)	
critical abnormalities (>75%)	14 (6.1)	15 (3.9)		3 (3.2)	3 (3.5)	
**COVID-19 infection severity indicators**						
Oxygen therapy flow rate of >5 L/min	172 (61)	247 (50.4)	**0.015**	67 (59.8)	55 (51.4)	0.26
Intubation/HNFO therapy/NIV	82 (29.1)	151 (30.8)	0.67	38 (33.6)	30 (26.5)	0.31
Intubation	80 (28.4)	140 (28.6)	1	38 (33.6)	26 (23)	0.10
HFNO therapy/NIV	11(3.9)	2 (0.4)	0.14	0	4 (3.5)	0.12
CT scan extension > 25% (N = 618)	111 (48.3)	188 (48.5)	1	0	4 (3.5)	0.12
CRP ≥ 100 mg/L (N = 746)	182 (65.7)	293 (62.5)	0.42	71 (64.5)	68 (60.2)	0.59
D-dimer count ≥ 1500 µg/L (N = 350)	99 (76.1)	151 (68.6)	0.16	48 (82.8)	35 (70)	0.18
Lymphopenia < 1000/µL (N = 753)	208 (174)	347 (73.5)	0.95	85 (75.9)	86 (77.5)	0.90
hs-cTnl ≥ 100 ng/L (N = 376)	37 (24.8)	40 (17.6)	0.12	14 (20.9)	12 (21.4)	1

BMI: Body mass index; COPD: Chronic obstructive pulmonary disease; CRP: C-reactive protein; CT: Computed tomography; eGFR: Estimated glomerular filtration rate; HFrEF: Heart failure with reduced ejection fraction; HFNO: High-flow nasal oxygen; hs-cTnl: High-sensitivity cardiac troponin; IQR: Interquartile range; M: Median; N: Number; NHF: Nasal high-flow; NIV: Noninvasive ventilation; PCR: Polymerase chain reaction; PS: Propensity score; RASi: Renin–angiotensin system inhibitor; VTE: Venous thromboembolism. * The *p*-value represents the difference between groups in the number of low-dose CT scans performed. Bold: significant *p* values and variables.

**Table 2 jcm-09-03472-t002:** In-hospital outcomes according to RASi treatment at baseline.

	Total N(%)/M(IQR)	RASi N(%)/M(IQR)	RASi-Free N(%)/M(IQR)	*p*-Value
**N**	**772**	**282**	**490**	
**Death**	173 (21.5)	82 (29.1)	91 (18.6)	**<0.001**
**Death/intubation**	322 (41.7)	133 (47.2)	189 (38.6)	**0.019**
**Death/intubation/HFNO therapy/NIV**	332 (43)	133 (47.2)	199 (40.6)	0.077
**Death/intubation/HFNO therapy/NIV/oxygen flow rate of ≥5 L**	438 (56.7)	179 (63.5)	259 (52.9)	**0.004**
**Death/intubation/HFNO therapy/NIV/oxygen flow rate of ≥10 L**	358 (46.4)	144 (51)	214 (43.7)	**0.047**
**Acute renal impairment** (N = 767)	212 (27.6)	105 (37.5)	107 (22)	**<0.001**
**Severe sepsis or septic shock** (N = 698)	132 (18.9)	55 (22.2)	77 (17.1)	0.103
**Pulmonary bacterial infection** (N = 747)	60 (8)	25 (9.2)	35 (7.4)	0.36
**Multiple organ deficiency** (N = 767)	36 (4.5)	16 (5.7)	20 (4.1)	0.31
**VTE**	61 (7.9)	21 (7.4)	40 (8.2)	0.72
**AF** (N = 748)	37 (4.9)	14 (5.2)	23 (4.8)	0.83
**Stroke**	20 (2.6)	10 (3.5)	10 (0.2)	0.21
**Major bleeding**	37 (4.8)	14 (5)	23 (4.7)	0.87
**Encephalitis**	15 (1.9)	6 (2.1)	9 (1.8)	0.86
**Liver injury** (N = 748)	18 (1.6)	5 (1.8)	13 (2.7)	0.60
**Hospital length of stay**	11 (6–21)	12 (7–24)	10 (6–20)	0.45
**Hospital length of stay of ≥30 days**	128 (16.6)	51 (18.1)	77 (15.7)	0.21

AF: Atrial fibrillation; HFNO: High-flow nasal oxygen; IQR: Interquartile range; M: Median; N: Number; NIV: Noninvasive ventilation; PCR: Polymerase chain reaction; RASi: Renin–angiotensin system inhibitor; VTE: Venous thromboembolism.

**Table 3 jcm-09-03472-t003:** Demographic, clinical, and paraclinical characteristics of the study population according to vital status at discharge.

	Total N (%)/m ± sd/ M(Q1–Q3)	Nonsurvivors N (%)/m ± sd/ M(Q1–Q3)	Survivors N (%)/m ± sd/ M(Q1–Q3)	*p*-Value
**N**	**772**	**173**	**599**	
**Age (years)**	68 (56–79)	79 (71–85)	65 (53–76)	**<0.001**
Age ≥ 65 years old	464 (60.1)	156 (90.2)	308 (51.4)	**<0.001**
**Male**	444 (57.5)	108 (62.4)	336 (56.1)	0.16
**BMI (kg/m^2^) N = 662**	28 (24–31)	27 (24–31)	28 (24–31)	0.10
**Cardiovascular risk factors**				
Hypertension N = 769	431 (56)	129 (74.6)	302 (50.4)	**<0.001**
Diabetes N = 769	219 (28.5)	62 (35.8)	157 (26.2)	**0.012**
Dyslipidemia N = 769	273 (35.5)	80 (46.5)	193 (32.3)	**<0.001**
Smoking (history or current) N = 671	155 (23.1)	49 (31.4)	106 (20.4)	**0.0042**
Obesity N = 690	241 (34.9)	50 (31.6)	191 (35.9)	0.32
**Medical history**				
Heart disease N = 768	124 (16.1)	37 (21.5)	87 (14.6)	**0.029**
Ischemic heart disease	91 (11.8)	58 (19.1)	33 (9.7)	**<0.001**
Chronic heart failure N = 769	43 (5.6)	21 (12.2)	22 (3.7)	**<0.001**
HFrEF	24 (3.1)	12 (6.9)	12 (2)	**<0.001**
Chronic kidney disease N = 769	112 (14.6)	55 (31.9)	57 (9.5)	**<0.001**
Chronic respiratory disease N = 769	91 (11.8)	26 (15.1)	65 (10.8)	0.13
COPD	45 (5.9)	14 (8.1)	31 (5.2)	0.14
Active cancer	53 (6.9)	26 (15)	27 (4.5)	**<0.001**
Cognitive impairment N = 768	95 (12.4)	46 (36.4)	49 (8.2)	**<0.001**
VTE N = 769	58 (7.5)	19 (11)	39 (6.5)	**0.048**
**Admission treatment**				
**Antithrombotic treatment**	267 (34.8)	99 (57.9)	168 (28.2)	**<0.001**
Antiplatelet N = 767	168 (21.9)	57 (33.3)	111 (18.6)	**<0.001**
Anticoagulation N = 766	118 (15.4)	52 (30.5)	66 (11.1)	**<0.001**
**Antihypertensive drugs**				
ACE or ARBs	282 (36.5)	82 (47.4)	200 (33.4)	**0.001**
Diuretics	176 (22.9)	69 (40.6)	107 (17.9)	**<0.001**
Beta-blockers	215 (28.1)	70 (41.2)	145 (24.3)	**<0.001**
**COVID-19 diagnosis**				
Positive PCR	772 (100)	173 (100)	599 (100)	
Low-dose CT	618 (80)	118 (68.2)	500 (83.5)	**<0.001 ***
normal	17 (2.8)	2 (1.1)	15 (2.5)	0.67 ^§^
uncertain abnormalities	12 (1.9)	6 (3.5)	6 (1)	
minimal (<10%)	84 (13.6)	9 (5.2)	75 (12.5)	
moderate (10–25%)	206 (33.3)	22 (12.7)	184 (30.7)	
important (25–50%)	162 (26.2)	36 (20.8)	126 (21)	
severe (50–75%)	108 (17.5)	33 (19.1)	75 (12.5)	
critical (>75%)	29 (4.7)	10 (5.8)	19 (3.2)	
**COVID-19 infection severity indicators**				
Oxygen flow rate of ≥5 L/min	378 (48.9)	135 (78)	243 (40.6)	**<0.001**
HFNO therapy/NIV/OTI	233 (30.2)	74 (42.8)	159 (26.5)	**<0.001**
OTI	220 (28.5)	71 (41)	149 (24.9)	**<0.001**
HFNO therapy/NIV	13 (1.7)	3 (1.7)	10 (1.7)	**1**
CRP ≥ 100 mg/L (N = 746)	476 (63.7)	133 (80.6)	343 (59)	**<0.001**
Ddimer count ≥ 1500 µg/L (N = 350)	250 (71.4)	72 (91.1)	178 (65.7)	**<0.001**
Lymphopenia < 1000/µL (N = 753)	555 (73.7)	147 (86.5)	408 (70)	**<0.001**
hs-cTnl ≥ 100 ng/L (N = 376)	77 (20.5)	37 (40.6)	40 (14)	**<0.001**
**In-hospital treatment**				
**Anticoagulant** N = 769	625 (81.3)	141 (82.4)	484 (80.9)	0.73
preventive	479 (62.2)	84 (49.1)	395 (66)	
therapeutic	145 (19)	56 (32.7)	89 (14.9)	
**ACEIs/ARBs**	125 (16.2)	28 (16.2)	97 (16.2)	1
**Antiviral**	286 (37)	55 (31.8)	231 (38.6)	0.65
lopinavir/ritonavir	153 (19.8)	29 (16.7)	124 (20.7)	
remdesivir	7 (0.9)	0	7 (1.2)	
oseltamivir	2 (0.5)	1 (0.6)	1 (0.2)	
hydoxychroloquine	140(18.1)	30 (17.3)	110 (18.4)	
IFN	4 (0.5)	0	4 (0.7)	
**Antibiotics**	634 (82.1)	141 (81.5)	493(82.3)	0.32
amoxicillin/clavulanic acid	239 (31)	31 (17.9)	208 (34.7)	
3GC	414 (53.6)	109 (63)	305 (50.9)	
macrolide	321 (41.6)	74 (42.8)	247 (41.2)	
quinolone	13 (1.7)	5 (2.9)	8 (1.3)	
other	3 (0.4)	2 (1.2)	1 (0.2)	

3GC: Third-generation cephalosporin; ACEI(s): Angiotensin-converting enzyme inhibitor(s); ARB(s) Angiotensin II receptor blocker(s); BMI: Body mass index; CI: Confidence interval; COPD: Chronic obstructive pulmonary disease; CRP: C-reactive protein; CT: Computed tomography; eGFR: Estimated glomerular filtration rate; HFrEF: Heart failure with reduced ejection fraction; hs-cTnl: High-sensitivity cardiac troponin; HFNO: High-flow nasal oxygen; HR: Hazard ratio; IFN: interferon; IQR: Interquartile range; M: Median; N: Number; NHF: Nasal high-flow; NIV: Noninvasive ventilation; OTI: orotracheal intubation; PCR: Polymerase chain reaction; RASi: Renin–angiotensin system inhibitor; VTE: Venous thromboembolism. * The *p*-value represents the difference between groups in the number of low-dose CT scans performed; ^§^ the *p*-value represents the difference between groups in the number of normal/abnormal low-dose CT.

**Table 4 jcm-09-03472-t004:** Univariate analysis of outcomes according to RASi pharmacological class (ACEIs versus ARBs).

Outcome	ACEIs *N* (%) *n* = 129	ARBs *N* (%) *n* = 152	Unadjusted OR ACEIs vs. ARBs (95%CI)	*p*-Value
Death	44 (34.1)	37 (24.3)	1.60 (0.95–2.71)	0.072
Severe pneumonia	86 (66.6)	92 (60.5)	1.30 (0.80–2.13)	0.288
Acute renal insufficiency	42 (32.6)	64 (42.1)	0.66 (0.40–1.08)	0.100

ACEI(s): Angiotensin-converting enzyme inhibitor(s); ARB(s) Angiotensin II receptor blocker(s); CI: Confidence interval; OR: Odds ratio.

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
