# Peer review of "Outcomes of COVID-19 Hospitalized Patients Previously Treated with Renin-Angiotensin System Inhibitors"

_jcm, 2020, doi:10.3390/jcm9113472_

Round 1

Reviewer 1 Report

I congratulate the authors on a well conducted retrospective case-control study evaluating the impact of RASi on COVID-19 outcomes.  The authors utilized a cohort of hospitalized patients in eastern region of France with 772 patients of which 431 had hypertension and 282 of which were on RASi.  The composite outcome of severe pneumonia and death were more frequently found in RASi patients though this association was not present after multivariate logistic regression analysis likely relating to increase severity of COVID-19 outcomes with hypertension along with other risk factors that were more represented in the RASi group (Table 1).  

The manuscript includes a helpful introduction outlining the interaction of RASi with ACE2 and its relevance in COVID-19.  The methods delineate the study design in detail and outline variables that were including in the evaluation. Univariate analysis was used to identify factors that independently associated with death and these were adjusted for in multivariate logistic regression analysis. After adjustment, age, active cancer, chronic kidney disease, and previous antithrombotic drug use were significantly associated mortality while age, male sex, CRP, and D-dimer were associated with severe pneumonia. Additionally, there was no difference between ACEi or ARB use (which were balanced in this cohort).

The study was well conducted however given the numerous retrospective analysis addressing the impact of RASi on COVID-19 infection rates and severity limits the originality of this manuscript.  It is important to note that there are limited studies that have looked at this question in a French cohort and could provide helpful data specific to this group. With a few modifications suggested below, a well conducted retrospective study could still add to the literature and be combined with other studies for meta-analysis to gain further confidence in the findings.

Suggestions:

  • The authors do a nice job of including a multivariable logistic regression analysis to identify the lack of association of RASi with COVID-19 severity. However, given the adequate size of the RASi and non-RASi groups, a propensity matching approach would provide a better control though this still will not account for hidden bias.
  • In table 2, the p-value for low-dose CT is unclear. Does that represent the p-value to determine if there is a difference between the number of patients in each group that received a low-dose CT or differences in the abnormalities found on CT? Similarly, it is unclear what the p-values in table 3 for COVID-19 diagnosis and medications.
  • it would be helpful under 2.3 Outcome Assessment to explicitly state that RASi medications were ACE inhibitors and Ang II receptor blockers (assuming renin inhibitors and mineralocorticoid receptor antagonists were not included).  I believe the assumption is that MRAs are not included. 
  • Were there any patients that were on RASi that did not have high blood pressure?  From Table 1 it seems there were 273 pts with hypertension that had RASI but 282 patients in the whole group were noted on RASi. Were the 9 patients without hypertension on RASi have kidney disease or heart failure? 
  • It would be helpful to note how many patients continued RASi after hospitalization. As there will be significant variability in dosage amongst patients and discontinuation during hospitalization, it will be difficult to run statistics on those patients. In line 197-198, - it is noted in-hospital continuation of ACEis/ARBS was not associated ith poorer outcome but this was not explicitly shown.
  • Line 237-238 – Increased ACE2 expression in the heart with hypertension and heart failure may not necessarily lead to higher risk of infection as viral entry is likely via eyes and nasopharynx.
  • Line 238 – It is not entirely clear that Age does not effect ACE2 expression. The following studies have suggested ACE2 expression varies with age in pediatric population (doi:10.1001/jama.2020.8707, doi.org/10.1016/j.jpeds.2020.08.037)
  • There were 282 patients that were noted in RASi group however under section 3.5 there are 129 pts on ACEis and 152 on ARBs which comes to 281 pts.  Do we know what the other patient was on?
  • How was lymphopenia defined?
  • Would refer angiotensin covertin enzyme 2 as ACE2 (line 261 referred as ACE-2
  • Suggest removing citation 30 "Mehra MR, Desai SS,.." and the references to it in the manuscript. Though it was noted to be retracted, the findings from the paper should not be discussed as the findings have been questioned.

Reviewer 2 Report

This is a retrospective study which basically confirms the accumulating notion that ACEI and ARB are of no harms in COVID19 patients. While this is probably true there are several issues that need attention. There were obvious differences in comorbidities between the patients who were and were not on RASIs. Obviously the patients with most of these comorbidities were on RASIs and this likely introduced a collinearity bias in their analysis, which did not seem to have been addressed properly.

In addition I have the following criticisms.

  1. The reasons for excluding 171 patients were not given and readers need to be reassured that no selection bias was introduced.
  2. The 2 populations (282 vs 490) examined were different from most of the parameters in a statistically significant way, for example, in the prevalence of HT or DM).
  3. It would have been better to compare the 282-RASi-users with matched  RASi-non-users hypertensive patients, also in the determination of their outcome (see Table 2), because their concomitant illnesses and baseline characteristics could have deeply affected the outcome results described in Table 2.
  4. As the RASIs + and RASIs - patients differed a propensity score matching should be considered and exploited.
  5. Available data from meta-analyses suggest that ACEIs can be protective and ARBs neutral. This did not emerge in this study likely due to a type 2 error. This should be discussed.
  6. Table 1: patients using RASi (282) are compared with patients not using RASi (490), a group that included both hypertensive and non-hypertensive patients.
  7. Instead there should be comparison between hypertensive patients with or without RASi). 
  8. References. One of the most interesting review on the topic of RASIs and their effect on the RAAS published in eLife at the beginning of the COVID 19 pandemics was overlooked. Quoting it might help in shortening the Discussion that is far too long and largely speculative,

Round 2

Reviewer 1 Report

I would like to congratulate the authors on completing a nice study evaluating the impacts of an important group of medicines (RASi) on COVID-19 outcomes form a French cohort.  Though the data is retrospective, it nicely correlates with additional data showing outpatient RASi use does not seem to impact COVID-19 outcomes in hospitalized patients. It is noted that 125 of the RASI group continued use during the hospitalization.  It would be helpful to analyze this group in comparison to the group that did not continue use in hospital though this was not analyzed but I think it would be outside the scope of this current manuscript.  

In line 301, I would clarify this Danish population look at prevalence of cardiac disease in COVID-19 cases.

The authors did a nice job incorporating propensity score.
